# The Enabling Effect of Digital Economy on High-Quality Agricultural Development-Evidence from China

**Junguo Hua [1], Jijie Yu [1], Yu Song [1,\*], Qi Xue [1] and Yujia Zhou [2,\*]**

1. College of Economics and Management, Henan Agricultural University, Zhengzhou 450046, China; hjghnnd168@163.com (J.H.); yjjhnnd@163.com (J.Y.); zhengzhouxueqi@163.com (Q.X.)
2. College of Business and Economics, Australian National University, Canberra 2600, Australia
* Correspondence: songyu@henau.edu.cn (Y.S.); zw05040712@163.com (Y.Z.)

**Abstract:** In recent years, the digital economy has shown great potential in regard to in driving social production and development. In the context of the construction of digital villages, the deep integration of the digital economy and agricultural development has injected new vitality into improving the quality and efficiency of agricultural production, becoming an important way to promote sustainable agricultural development. Based on the panel data of 31 provinces in China from 2012 to 2021, the study utilizes the entropy method to measure the level of the digital economy and the high-quality development of agriculture. Additionally, this study explores the impact and mechanism of the digital economy on the high-quality development of agriculture by the fixed effect, mediation effect, and the spatial spillover models. In summary, the digital economy can significantly drive the high-quality development of agriculture, which is still valid after considering endogeneity and robustness. Mechanistically, the rationalization of industrial structure is an important path for the digital economy in regard to driving the high-quality development of agriculture. Regionally, the dividends of the digital economy for high-quality agricultural development in the central and western regions are greater than those in the eastern region. Spatially, the digital economy has a spatial spillover effect on the high-quality development of agriculture. Moreover, it can promote the synergistic development of adjoining regions. Therefore, policy recommendations are made in terms of strengthening rural infrastructure, emphasizing the development of regional shortcomings, and strengthening internal with external regional linkages.

**Keywords:** digital economy; high-quality agricultural development; industrial structure rationalization; sustainable development

## 1. Introduction and Literature Review

High-quality agricultural development is a new development model for agriculture that is driven by innovation to improve the quantity and quality of agricultural products, and it coordinates the rural industry with the urban and rural structures, leads low-carbon development in agriculture with greenery, opens up and optimizes agricultural resources and markets, and shares the fruits of development with farmers [1]. Agriculture, as a pillar industry of the national economy, is the basis for realizing economic and social development and acts as a key force that is indispensable in terms of driving sustainable development. However, at the current stage of development, China's agriculture is still facing a series of challenges, such as insufficient endogenous dynamics, low digital literacy of farmers, irrational industrial structure, and low resource utilization [2]. To solve these problems and achieve high-quality agricultural development, it is imperative that we continue to push forward the reform of agricultural supply, depend on modern science and technology, and use science and technology to promote agriculture. The digital economy is a new opportunity for China's economic development, which can realize the organic combination of innovation elements, innovation subjects, and innovation

links between different fields through the use of advanced digital technology, promoting the reform of the rural industry, farmers' lives, rural governance, etc. while continuing to release development dividends to the agricultural field and provide new impetuses for the upgrading of the rural industrial structure [3]. Based on the above, it is very important to explore the mechanism of the impact of the digital economy on the high-quality development of agriculture to promote the strategy of rural revitalization and achieve sustainable agricultural development in China.

The quality of agriculture is reflected in the development model of greening, quality, specialization, and branding of agriculture, which aims to achieve the enhancement of the competitiveness of agricultural products in the market and the upgrading of the agricultural industry chain while emphasizing the high degree of coupling and coordination between agriculture and the ecological environment, society, and humanities [4]. For the connotation and realization path of high-quality agricultural development, academics have currently conducted in-depth research from multiple perspectives and used different methods to measure it. Regarding the connotation of development, Xia Xianli et al. put forward that high-quality agriculture is an organic whole composed of an industrial system, production system, and operation system [5]. Wang Xingguo et al. think that high-quality agriculture is a development model which aims at satisfying people's increasing demands for a better life, using "Five Developmental Concepts" and "Quality and Efficiency" as the guiding principles [6]. In terms of the realization path, the incremental input of resource factors such as land and labor is the most common explanation [7]. However, it is very difficult to develop agriculture with the help of factor increment alone, and technological innovation and digital finance are the important factors that influence high-quality agricultural development [8,9]. In terms of measuring and evaluating the quality of agricultural development, Barras et al. used a simple indicator to measure the quality of agricultural development in Slovenia [10]. Additionally, Yang Junge and his colleagues, based on the new development idea, have made a comprehensive evaluation of the quality of the agricultural development system utilizing the multi-dimensions index [11]. The above research provides the theory support for the research of high-quality agriculture, but the relationship between the digital economy and high-quality agricultural development needs to be further explored.

Regarding the research on digital economy empowering high-quality agricultural development, academics mainly explore the impact relationship, constraints, and realization path. In the relationship between the two influences, the embedding and application of digital economy in the field of agricultural operation can improve agricultural output, realize accurate prediction and control of agriculture, and drive the transformation of traditional agriculture to smart agriculture [12–14]. In addition, the digital economy gradually drives the flow of capital, technology, and talent to the agricultural aggregation with the flow of information, promoting the development of traditional agriculture to smart agriculture and digital agriculture through the recombination of agricultural production factors. Using cluster analysis, Song et al. concluded that the digital economy can optimize the methods of agricultural operation and management, improve peasants' market coupling ability, and then promote agricultural industry transitions and upgrades [15]. At present, the process of agricultural digital transformation is accelerating and certain achievements have been made; however, there are still constraints. China has many practical dilemmas, such as high dependence on foreign countries for core technologies, high cost of production factors, and insufficient soft power for international competition [16,17], thus presenting weak digital transformation capabilities, unsound infrastructures, imperfect data sharing mechanisms, and low technological convergence in the field of agriculture [18]. For this reason, based on the needs of the national economic strategy, the construction of a high-level digital industrial system has a significant driving value in promoting the modernization and sustainable development of Chinese agriculture. As far as the realization path is concerned, the digital economy can help traditional industries develop toward quality by promoting technological innovation, improving production processes, and reducing production costs [19]. Ozili

found that the digital economy promotes the free flow and allocation of capital elements in the financial market, creating a favorable market environment for financial inclusion [20]. By providing efficient and precise financial services for agricultural business entities, it breaks through the limitations of financing and optimizes the allocation of agricultural resources, thus enhancing the efficiency of agricultural production [21]. At the same time, several scholars have conducted empirical research on the relationship between the digital economy and rural agriculture through provincial panel data, finding that it can promote the high quality of agriculture by employing technology innovations, human capital quality, and industrial structure change, providing empirical evidence in this area [22–24].

To summarize, although the current research on issues related to digital and agricultural development is quite rich, there is still a lot of room for the digital economy to promote high-quality agricultural development. Compared with the existing literature, the marginal contribution of this study is as follows: First, based on the agricultural perspective, it constructs the index system of agricultural high-quality development from the four dimensions of power enhancement, quality change, structural optimization, and green development, and it also comprehensively tests the empowering effect of the digital economy on agricultural high-quality development from three aspects of the direct, mediating, and spatial spillover effects, which is theoretically valuable for expanding the field. Second, from the angle of agriculture development, it is helpful to clarify the methods of high-quality agriculture development and also to study the mechanism of the influence of digital economy on the quality of agriculture. Third, from the point of view of research value, the study enriched and supplemented relevant research on digital agricultural development and provided a reference for precise policymaking for high-quality agricultural development.

The above section of the study provides the introduction and literature review, and the remaining section is structured as follows: Section 2 presents the theoretical analysis and research hypotheses. Section 3 describes the research design in terms of model construction, variable measurement, and data description. Section 4 reports the empirical test results and analysis. Section 5 summarizes the findings of this study and suggests countermeasures. Section 6 provides a discussion. Section 7 presents the limitations of this study.

## 2. Theoretical Analysis and Research Hypothesis

### 2.1. Analysis of the Direct Effects of the Digital Economy on High-Quality Agricultural Development

High-quality development in agriculture is a new model for studying the development of the agricultural sector from the point of view of agriculture, and it plays an important role in China's efforts to enhance scientific and technological innovation in agriculture, raise the level of quality and safety of agricultural products, promote farmers' incomes, and promote sustainable development [25]. In recent years, China has successively formulated sevaral policies and measures to promote high-quality development in agriculture. Although there are differences in the interpretation and definition of the connotation of high-quality development in agriculture in different fields, these measures are ultimately implemented in the four aspects of increasing production efficiency, improving product quality, optimizing industrial structure, and promoting green environmental protection [23]. On this basis, this study analyzes the four dimensions of agricultural power improvement, agricultural quality change, agricultural structure optimization, and agricultural green development.

According to the new economic growth theory, technology development is a powerful impetus for the economy, particularly in the era of the digital economy, where the generation, dissemination, processing, and utilization of information have a profound impact on market behavior and economic efficiency [26]. The emergence of digital technology has brought great changes to the agricultural sector. By establishing a data synergy system, the digital economy powerfully breaks the opacity of information between upstream and downstream, enabling agricultural producers to rapidly access key market information, meteorological data, agricultural technology, etc., thus providing strong technical support for advanced agriculture [27]. Firstly, the digital economy can promote the improvement

of agricultural power. China's traditional agriculture has typical "small-scale, part-time" characteristics, such as agricultural material prices, and the labor costs caused by the rise of agricultural production marginal benefits are low, limiting the development of agricultural modernization. As the digital economy deepens, many new techniques have been developed in the production and sale of agricultural products, which has greatly reduced the cost of agricultural products and strengthened the driving force of agriculture. Secondly, the digital economy is contributing to quality change in agricultural development. As digital technology develops and is applied, China's agricultural informatization is being continuously improved, making more accurate and efficient use of agricultural resources. Through the scientific evaluation of meteorological elements, it can reduce losses caused by natural disasters, stabilize agricultural production, improve the quality of agricultural production, and promote sustainable agricultural development. In addition, through electronic monitoring of agricultural conditions, sensors, and data analysis, scientific management of arable land has been realized, guiding farmers to carry out scientific and rational agricultural production. Thirdly, the digital economy is beneficial in regard to optimizing the structure of agriculture. In the traditional distribution of agricultural products, there are intricate distribution channels, from production to sales, often through multiple levels, which inevitably results in longer transportation time, higher prices, and a loss of quality in agricultural products. With the advancement of digital technology, traditional production and distribution methods will also be transformed into intelligence, the sales process of agricultural products will be simpler and more transparent, the coverage of the market will be larger, and the scattered consumer demand will be quickly aggregated on the e-commerce platform, which will achieve an accurate match between supply and demand, thus promoting the high-quality development of agriculture. Finally, the digital economy can promote the greening of agricultural production. On the one hand, digital elements are characterized by low marginal costs, as well as intangible services, and sharing, which gives digital agriculture the characteristics of high energy, low consumption, and cleanliness, which are different from those of agriculture, manufacturing, and traditional services. Digital agriculture can not only improve labor productivity and the added value of agricultural products but can also reduce resource consumption and improve cleanliness. On the other hand, with the digital transformation of traditional industries, all kinds of production and operation data are connected to the big data platform through the Internet, cloud computing, and other media, and supervisory authorities supervise through data visualization and other means to promote the greening of agricultural production.

**Hypothesis 1.** *The digital economy is a significant contributor to the high quality of agriculture.*

### 2.2. Analysis of Mediating Effects of the Digital Economy on High-Quality Agricultural Development

The theory of innovation makes it clear that technological change and the development of new products and services are important drivers of economic growth and the upgrading of the industrial structure. The advanced production methods brought about by digital technology have led to a more optimal allocation of resources and a tendency for the weight of different industries to become more advanced. Agricultural production and operation take advantage of the development of the information and communication industry and its spillover and diffusion effects on other industries in order to drive the concentration of talent, technology, and capital flows in the field, promote the rationalization of the agricultural industrial structure, and achieve high-quality development of agriculture [28].

Firstly, the digital economy can promote the rationalization of the agricultural business model. Information technology can help agricultural business entities form new business consortia. Agricultural cooperative organizations linked by digital platforms can establish an online agricultural industry cluster on the network so that various elements within it complement each other's strengths and develop synergistically. In addition, rural e-commerce can break the limitations of time and space, replace the traditional multi-level distribution with direct sales at the place of origin, reconstruct the supply chain and value

chain of agricultural products, and realize management networking. By developing and applying large data to consumers, we can adjust the structure of agriculture in time and achieve the exact matching of the demand and supply. Secondly, the digital economy can promote the rationalization of the internal structure of agriculture. The development of the digital economy can help change traditional agricultural production methods, promote its development in the direction of standardization, and improve the segmentation of agricultural industries such as crop cultivation, livestock breeding, fishing, hunting, and aquaculture. With the help of big data, digital technology can fully analyze the characteristics and connections between different industries, thus forming a new type of agricultural industry chain. On the one hand, it can explore the value of original products and promote the development of cold products. In practice, it will also provide new ideas and methods to promote synergy and integration in Chinese agriculture [29]. Finally, the digital economy can promote the rationalization of the external structure of agriculture. This is mainly reflected in the fact that the digital economy can promote the extension of the industrial chain and strengthen the link between agriculture and secondary and tertiary industries [30]. The development of the actual economy is conducive to breaking down the barriers between industries, establishing an agricultural digital sharing platform, helping to ease the reality of the dilemma of information asymmetry between industries, realizing the digital integration of agriculture and other industries in the production, processing, and marketing system, and promoting the integration of agricultural production, supply, and marketing. In addition, the digital economy has weakened the drawbacks of traditional agricultural production and business transactions, and the link between agriculture and the service industry will become closer. Farmers can use the service industry model for product sales, increasing the income of rural people while realizing the improvement of people's living standards.

**Hypothesis 2.** *The digital economy can significantly contribute to high-quality agricultural development through the rationalization of industrial structures.*

*2.3. Analysis of the Spatial Effects of the Digital Economy on High-Quality Agricultural Development*

According to the theory of spatial econometrics, the flow, diffusion, and spillover of resource factors will enhance the spatial dependence among economies. The digital economy facilitates the collection, processing, analysis, and handling of data, compresses the distance of information transmission between different regions, and enhances the connection of economic activities between regions. First of all, the widespread use of information technology means that space is no longer a limiting factor in regard to economic activities, and it has also accelerated the flow of labor and capital and weakened the correlation between economic activities and geographic location, which is an important factor in strengthening interpersonal, regional, and urban-rural ties [31]. Some regions have made significant progress in socio-economic development, thanks to high-quality digital talent and improved infrastructure, while also making the level of advanced production spatially uneven. Secondly, digital technology constantly innovates the mechanism mode of economic exchange and cooperation, introduces new production factors and combinations into the production system to create new value, and strengthens the information exchange of interregional agricultural market players [32]. The cross-regional flow and integration of factors have improved the governance capacity of China's agricultural economy, thus realizing the optimal allocation of agricultural resources and improving agricultural productivity. Finally, the digital economy does not develop in isolation, and its expansive nature continues to contribute to its geographic expansion and interaction so that its development has some spatial relevance; therefore, its impact on the quality of agricultural development will potentially also have spatial implications.

**Hypothesis 3.** *The enabling role of the digital economy for high-quality agricultural development has spatial spillover effects.*

Based on the above theoretical analysis, the study of the impact mechanism of digital economy in this paper is shown in Figure 1, which contains the direct impact of the digital economy on the high-quality development of agriculture, as well as the possible indirect impact of the rationalization of the industrial structure.

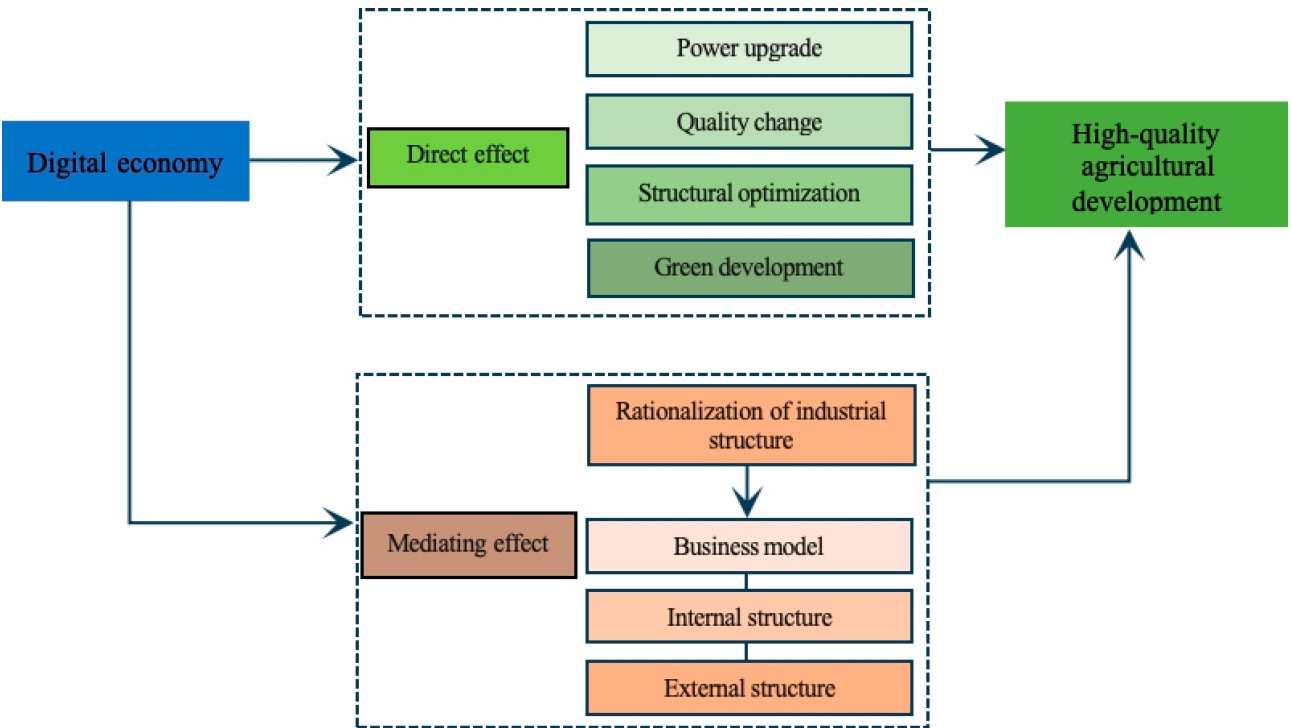

**Figure 1.** Mechanisms of action of the digital economy in influencing the high-quality development of agriculture.

## 3. Methodology and Data

### 3.1. Equation Specification

To examine the influence of digital economy on high-quality agricultural development, the study constructs the following basic model:

$$HQAD_{it} = \alpha_0 + \alpha_1 DE_{it} + \alpha_2 Z_{it} + \varepsilon_{it} + \mu_i + \nu_t \tag{1}$$

where $HQAD_{it}$ is the level of high-quality agricultural development in region *i* in year *t*, $DE_{it}$ is the level of digital economy development in region *i* in year *t*, $Z_{it}$ is a control variable that may influence the level of high-quality agricultural development, $\varepsilon_{it}$ is a random perturbation term, $\mu_i$ is an individual fixed effect, and $\nu_t$ is a time fixed effect.

To examine the role of the digital economy in promoting high-quality agricultural development, this study refers to the method of Wen Zhonglin et al. to build a mediation effect test model [33] to test the role of the path of the rationalization of industrial structure:

$$M_{it} = \beta_0 + \beta_1 DE_{it} + \beta_2 Z_{it} + \varepsilon_{it} + \mu_i + \nu_t \tag{2}$$

$$HQAD_{it} = \gamma_0 + \gamma_1 DE_{it} + \gamma_2 M_{it} + \gamma_3 Z_{it} + \varepsilon_{it} + \mu_i + \nu_t \tag{3}$$

*M* in the above formula indicates the rationalization of industrial structure. Equation (2) mainly studies the effect of digital economy on the mediating variable *M*. Equation (3) studies the effect of digital economy and mediating variable *M* together on the high-quality development of agriculture.

### 3.2. Variable Measurement

3.2.1. Dependent Variable

In this study, the level of high-quality agricultural development was selected as the dependent variable (HQAD). High-quality agricultural development is a comprehensive development concept aimed at realizing high-yield, high-quality, and high-efficiency agricultural production while promoting the sustainable development of the agricultural economy. This definition emphasizes the importance of the green development concept to enhance the quality and market competitiveness of agricultural products while increasing farmers' incomes and realizing the coordinated development of the economy, society, and environment. Based on the connotation of high-quality development of agriculture and the research of Yi Enwen and Wang Jin [34,35], this study constructs the index system from the four dimensions of power enhancement, quality change, structural optimization, and green development.

Considering that the entropy value method can give different weights according to the entropy value of the indicators, to weight and sum different indicators, avoiding the influence of subjective factors. At the same time, the entropy value method can also eliminate the influence of correlation by correlation analysis of the indicators and obtain a comprehensive evaluation result. Therefore, this paper utilizes the entropy method to measure the weights of each index, and we obtain the index of high-quality development of agriculture for the 31 provinces during the period of 2012–2021. Specifically, power improvement includes mechanization level, land productivity, labor productivity, and rural economic efficiency. The quality change includes technical support for agriculture, the rural marketization level, and the industrial integration level. Structural optimization includes the ratio of non-farm employment in the countryside, industrial structural adjustment, and urban-rural dichotomy. Green development includes the extent of chemical fertilizer inputs, the extent of agricultural inputs, the amount of agricultural plastic film used, as well as the ratio of effective irrigated area. All indexes and the corresponding calculation methods are expressed in Table 1.

**Table 1.** System of indicators for the level of high-quality development of agriculture.

| First Index | Secondary Index | Index Calculation Method | Direction | Weights |
|---|---|---|---|---|
| Power upgrade | Degree of mechanization | Total power of agricultural machinery/total area under cultivation | + | 0.1365 |
| | Land productivity | Grain production/area sown with grain | + | 0.0505 |
| | Labor productivity | Gross output value of agriculture, forestry, animal husbandry, and fishery/number of persons employed in the primary sector | + | 0.0909 |
| | Rural economic benefits | Per capita disposable income of rural residents | + | 0.0888 |
| Quality change | Agricultural technical support | Number of persons engaged in agricultural scientific and technological activities/number of persons employed in the primary sector | + | 0.1954 |
| | Level of rural marketization | Consumption expenditure per rural inhabitant | + | 0.0653 |
| | Level of industrial integration | Output of agriculture, forestry, and fishery services/total output of agriculture, forestry, and fishery services | + | 0.1001 |
| Structural optimization | Percentage of rural non-farm employment | 1—(Number of rural workers in primary sector/rural population) | + | 0.0578 |
| | Industrial restructuring | Value added of primary sector/GDP | − | 0.0293 |
| | Urban-rural dichotomy | Ratio of per capita income of urban and rural residents | − | 0.0261 |
| | | Ratio of per capita consumption of urban and rural residents | − | 0.0131 |

**Table 1.** *Cont.*

| First Index | Secondary Index | Index Calculation Method | Direction | Weights |
|---|---|---|---|---|
| Green development | Fertilizer input intensity | Fertilizer application/total agricultural output | − | 0.0149 |
| | Pesticide input intensity | Pesticide use/gross agricultural output | − | 0.0153 |
| | Agricultural plastic film use | Rural plastic film use/gross agricultural product | − | 0.0110 |
| | Effective irrigated area ratio | Effective irrigated area/total sown area of crops | + | 0.1052 |

### 3.2.2. Key Independent Variable

This study takes digital economy as the key independent variable (DE). The digital economy mainly uses information technology, such as digitization, networking, and intelligence, to increase the efficiency of agricultural production, empower the high-quality development of agriculture, and have a positive empowering influence on national economic growth, industrial conversion, and upgradation, reducing carbon emissions, improving the social governance system, and reducing air pollution [36–40]. At present, the research has not yet formed a set of generally recognized evaluation index systems for the digital economy. Under the principles of scientificity and comprehensiveness, the study refers to the research results of related literature and the way some scholars select indicators for the digital economy [41,42], establishing a digital economy indicator system from the three aspects of the digitalized transaction basis, digitalized industry application, and digital technology innovation. The detailed composition of the indicators is expressed in Table 2, and the entropy value method was used to calculate the weight of every indicator, while the digital economy development index was calculated and recorded as DE.

**Table 2.** Indicator system for the level of development of the digital economy.

| First Index | Secondary Index | Index Calculation Method | Direction | Weights |
|---|---|---|---|---|
| Fundamentals of digital trading | Traditional infrastructure | Internet broadband access port | + | 0.0510 |
| | | Number of IPv4 addresses | + | 0.1188 |
| | New infrastructure | Length of long-distance fiber optic cable lines | + | 0.0313 |
| | | Cell phone penetration rate | + | 0.0219 |
| Digital industry applications | Digital industrialization | Total telecommunication services | + | 0.1040 |
| | | E-commerce sales | + | 0.1181 |
| | Industrial Digitization | Websites per 100 businesses | + | 0.0087 |
| | | Number of degrees awarded | + | 0.0360 |
| Digital technology innovation | Innovation environment | Number of patent applications granted | + | 0.1245 |
| | | R&D expenditures of industrial enterprises | + | 0.1046 |
| | Innovation outputs | Revenue from sales of new products | + | 0.1175 |
| | | Technology market turnover | + | 0.1636 |

### 3.2.3. Mediating Variable

In this study, industrial structure rationalization (ISR) is used as a mediating variable that reflects the synergistic ability and the degree of correlation between industries. Referring to the study of Chunhui Gan et al. [43], this study selects the Thiel index to measure industrial structure rationalization (ISR). The calculation method of the Thiel index is shown in Equation (4):

$$TR = \sum_{j=1}^{m} S_i S_j \frac{X_{ij}}{X} ln \frac{X_{ij}}{X} \qquad (4)$$

In Equation (4), *TR* represents the Theil index, *i* represents the region, *j* represents the industry, $S_i$ denotes the proportion of the population of region *i* to the total population of the country, $S_i$ denotes the ratio of the output value of the *j*th industry to the total output value of the country, $X_{ij}$ denotes the per capita output value of the *j*th industry of region *i*, and *X* denotes the total output value per capita of the country.

### 3.2.4. Control Variables

After identifying the main explanatory variables, the paper also needs to control for other factors that may make an impact on high-quality agricultural development, with the following variables being chosen as control variables: (1) economic environment (ECON), which is expressed as per capita GDP in logarithmic form for each region; (2) urbanization rate (UR), which is measured by the ratio of the civic resident population to the total population of each region; (3) human capital loss (EDR), which is measured by the dependency ratio of the elderly population; (4) agriculture structure upgrading (AUP), which is measured using the ratio of the annual operating income of leisure agriculture to the total output value of the primary industry; and (5) the government support for agriculture (GSA), which is measured using the expenditure on the agricultural, forestry, and water affairs of the local finances in logarithmic form. Table 3 lists the abbreviations, meanings and measurements of all continuous variables.

**Table 3.** Description of the variables.

| Variable | Acronym | Basic Meaning | Calculation Method |
|---|---|---|---|
| Dependent variable | HQAD | High-quality agricultural development | Entropy method |
| Core explanatory variable | DE | Digital Economy | Entropy method |
| Mediating variable | ISR | Rationalization of industrial structure | Thiel index |
| Control variable | ECON | Economic environment | GDP per capita by region in logarithmic form |
| | UR | Urbanization rate | Share of urban resident population in the total population of each region |
| | EDR | Human capital loss | Elderly dependency ratio |
| | AUP | Agriculture structure upgrading | Share of annual operating income from leisure agriculture in the output value of the primary sector |
| | GSA | Government support for agriculture | Expenditures on agriculture, forestry, and water affairs in local finance in logarithmic form |

### 3.3. Data Description

The study sets the research period as 2012–2021 and selects 31 provinces and cities in mainland China as the research subject. The relevant data mainly come from the China Statistical Yearbook, China Agricultural Statistical Yearbook, China Science and Technology Statistical Yearbook, and the statistical yearbooks of each province, as well as the Cathay Pacific database and EPS database. The descriptive statistics of the variables are expressed in Table 4. Next, this study will utilize the econometric analysis method to further rigorously analyze the relationship between the digital economy and the high-quality development of agriculture in a quantitative manner.

**Table 4.** Descriptive statistics of the variables.

| Variable | Obs. | Mean | Std. Dev. | Min. | Max. |
|---|---|---|---|---|---|
| HQAD | 310 | 0.3040 | 0.104 | 0.12 | 0.58 |
| DE | 310 | 0.1264 | 0.112 | 0.02 | 0.55 |
| ISR | 310 | 2.4068 | 0.118 | 2.20 | 2.82 |
| ECON | 310 | 10.8997 | 0.428 | 10.04 | 12.01 |
| UR | 310 | 0.5932 | 0.127 | 0.29 | 0.89 |
| EDR | 310 | 0.1517 | 0.042 | 0.08 | 0.25 |
| AUP | 310 | 0.4854 | 0.093 | 0.33 | 0.81 |
| GSA | 310 | 0.1164 | 0.034 | 0.04 | 0.19 |

Note: Table 4 lists the descriptive statistics of the variables, including sample size, mean, standard deviation, and maximum value, and the data were indented by 1% through Stata 16.0 software.

## 4. Empirical Testing and Discussion

### 4.1. Benchmark Regression

To estimate equation (1), as constructed in the previous study, and to test the direct effect of DE on HQAD, this study conducts a series of empirical tests, with the results being presented in models (1)–(5) in Table 5. Considering the comparative analysis, as well as the robustness of the models, the study uses the least squares (OLS) model and the individual fixed effects model (FE) for comparison. Models (1) and (2) are the results of the mixed OLS regression model both with the control variables removed and with the control variables, respectively. Models (3), (4), and (5) are fixed effects models without control variables, with individuals only, and with control variables added, respectively. It is then found that the coefficients of DE are all significant, and the results are still significant and positive at the 1% level after adding control variables and controlling for the effects of province and year, which suggests that DE can enhance HQAD. Hypothesis 1 is confirmed. In addition, the two-way fixed-effects model can take into account individual heterogeneity and temporal heterogeneity, which can more accurately estimate the effect of the independent variable on the dependent variable, and the results of the Hausman test indicate that the FE model is a more preferable choice, so the study chooses model (5) for analysis. In this case, the impact coefficient of DE is 0.177, which passes the test of significance at a 1% level, indicating that, for every unit increase in DE, HQAD will be increased by 17.7% and Hypothesis 1 is confirmed again. From model (3) and model (5), it can be seen that the regression coefficient of DE decreases from 0.293 to 0.177 after adding control variables, which indicates that disregarding the control variables will exaggerate the driving function of DE. Finally, as for control variables, there is a positive and significant relationship between economic environment, agricultural structure upgrading, and agricultural quality development, indicating that a good economic environment and agricultural structure upgrading can help to enhance the deep of agricultural quality development. There is also a negative correlation between the urbanization rate, financial support for agriculture, and high-quality agricultural development, which may be attributed to the changing of advanced rural labor to the cities and the irrational structure of agricultural trade that affects the degree of high-quality agricultural development.

### 4.2. Endogeneity and Robustness Tests

#### 4.2.1. Endogeneity Test

The digital economy can drive high-quality agriculture development, but there may be endogeneity problems between the two. This study intends to use the instrumental variable method and independent variable lagged one-period treatment to reduce the endogeneity problem produced by omitted variables and two-way causality.

**Table 5.** Benchmark regression results.

| Variable | OLS | | | FE | |
|---|---|---|---|---|---|
| | **(1)** | **(2)** | **(3)** | **(4)** | **(5)** |
| DE | 0.628 *** | 0.161 *** | 0.293 *** | 0.214 *** | 0.177 *** |
| | (16.30) | (3.89) | (8.91) | (6.10) | (5.05) |
| ECON | | 0.213 *** | | 0.120 *** | 0.089 *** |
| | | (12.10) | | (10.00) | (7.03) |
| UR | | −0.384 *** | | 0.032 | −0.184 ** |
| | | (−6.34) | | (0.42) | (−2.27) |
| EDR | | −0.026 | | 0.337 *** | −0.065 |
| | | (−0.31) | | (4.79) | (−0.76) |
| AUP | | 0.277 *** | | 0.174 *** | 0.145 *** |
| | | (5.93) | | (4.13) | (3.07) |
| GSA | | −0.333 ** | | −0.395 *** | −0.487 *** |
| | | (−2.44) | | (−3.83) | (−4.81) |
| Constant | 0.225 *** | −1.902 *** | 0.212 *** | −1.135 *** | −0.621 *** |
| | (34.48) | (−11.52) | (50.20) | (−11.02) | (−4.57) |
| Province effect | | | Yes | Yes | Yes |
| Time effect | | | Yes | No | Yes |
| N | 310 | 310 | 310 | 310 | 310 |
| $R^2$ | 0.463 | 0.730 | 0.881 | 0.888 | 0.907 |

Note: ** and *** represent 5% and 1% significance levels, respectively, and the value in () is the t value.

First, the instrumental variable method. Referring to the practice of previous literature, the digital economy with one period lag is chosen as an instrumental variable. This variable has a strong correlation with the digital economy in the current period, and the model (6) in Table 6 shows the test results of introducing the instrumental variable. It can be found that DE passes the significance test at the 1% statistical level, which is consistent with the results of the benchmark regression. In addition, the Cragg- Donald Wald F statistic is 760.18, which is greater than the critical value of Stock Yogo's weak instrumental variable of 16.38. The Kleibergen Paap rk LM value is 7.03, which rejects the original hypothesis of non-identifiability at the 1% level. In conclusion, the driving effect of DE on HQAD remains after considering endogeneity, and hypothesis 1 is further tested.

**Table 6.** Robustness test results.

| Variable | (6) | | (7) | (8) | (9) |
|---|---|---|---|---|---|
| | **Phase I** | **Phase II** | **One Period Behind** | **Adjustment Range** | **Exclusion Sample** |
| DE | | | 0.205 *** | 0.240 *** | 0.229 *** | 0.231 *** |
| | | | (3.35) | (6.29) | (5.53) | (6.42) |
| Control variable | Yes | | Yes | Yes | Yes | Yes |
| Instrumental variable | 0.964 *** | | | | | |
| | (6.77) | | | | | |
| Constant | 0.008 | | −0.563 ** | −1.085 *** | −0.421 ** | −0.549 *** |
| | (0.06) | | (−2.25) | (−9.69) | (−2.50) | (−3.69) |
| Province effect | YES | | YES | YES | YES | YES |
| Time effect | YES | | YES | YES | YES | YES |
| Cragg-Donald Wald F | | 760.18 {16.38} | | | | |
| Kleibergen-Paaprk rk LM statistic | | 7.03 {0.008} | | | | |
| N | 279 | | 279 | 279 | 217 | 270 |
| $R^2$ | 0.963 | | 0.979 | 0.881 | 0.882 | 0.908 |

Note: ** and *** represent 5% and 1% significance levels, respectively.

Second, the independent variables are lagged by one period. Considering that DE has a certain time lag, the study lags the independent variables by one period and conducts regression. The consequences of the model (7) in Table 6 show that DE can still significantly promote HQAD, while as the empirical results are consistent with the above.

4.2.2. Robustness Test

To avoid chance in the generation of regression results, the study conducts robustness tests by adjusting the sample interval as well as eliminating the special sample method to enhance the stability and reliability of the study's conclusions.

First, the sample interval is adjusted. China's digital economy development has been encountering a booming period since 2015, and it is more meaningful to focus on examining the level of digital economy development during the boom period [44]. This paper will examine with the case of Liu Jun et al. to adjust the sample interval to 2015–2021 to regress, and the specific regression consequences are revealed in Table 6 Column (8), the coefficient of DE is still significantly positive, indicating that the findings of this paper are reliable.

Second, special samples are excluded. Due to the strong economic strength and influence of Beijing, Tianjin, Shanghai, and Chongqing, the data from the four cities are eliminated in this paper and the samples are returned. Table 6 model (9) shows that the direction of the function of DE on HQAD is consistent and significant with the above results, which proves that the research results are reliable.

*4.3. Tests for Mediation Effects*

To estimate Equations (2) and (3), constructed in the previous section, and to test Hypothesis 2, this paper empirically tests the mediating effect of ISR between DE and HQAD, with the results being shown in Table 7. Model (10) reports the effect of DE on HQAD, model (11) is the effect of DE on ISR, and model (12) reports the results of the simultaneous regression of DE and ISR. In model (10), the measured coefficient of DE is positive and significant, which indicates that DE has a positive contribution to HQAD. In model (11), the coefficient of DE is significantly positive, which indicates that DE has a facilitating effect on ISR. In model (12), the coefficients of both DE and ISR are significantly positive, and the coefficient of DE is reduced relative to model (10). It can be seen that DE can promote HQAD through the mediating effect of ISR, and ISR has a mediating effect between DE and HQAD, which shows the transmission mechanism of "DE-ISR-HQAD". In the context of the digital economy, big data information technology can be used to analyze the characteristics and relationships between different industries. Meanwhile, it can also facilitate the rational changing of the structure of the agricultural industry through integration and innovation, thus promoting the high-quality development of agriculture.

**Table 7.** Mechanism test results.

| Variable | Models (10) | Models (11) | Models (12) |
|---|---|---|---|
| | HQAD | ISR | HQAD |
| DE | 0.1767 *** | 0.0781 ** | 0.1616 *** |
| | (5.0549) | (2.4374) | (4.6367) |
| ISR | | | 0.1932 *** |
| | | | (2.9174) |
| Control variable | Yes | Yes | Yes |
| Constant | −0.6208 *** | 1.9276 *** | −0.9933 *** |
| | (−4.5739) | (15.5002) | (−5.3700) |
| Province effect | Yes | Yes | Yes |
| Time effect | Yes | Yes | Yes |
| N | 310 | 310 | 310 |
| $R^2$ | 0.892 | 0.869 | 0.895 |

Note: ** and *** represent 5% and 1% significance levels, respectively.

### 4.4. Heterogeneity Analysis

The degree of development of Chinese provinces is inconsistent, resulting in the inevitable regional differences in the level of development of digital agriculture. As a result, this study divides the study sample into three regions: east, central, and west. The regression results are shown in Table 8.

**Table 8.** Results of heterogeneity test.

| Groups | Region | HQAD | Obs. | Control Variables |
|--------|--------|------|------|-------------------|
| DE | Eastern part | 0.0606<br>(1.2677) | 110 | Yes |
| | Central part | 0.9935 ***<br>(6.7760) | 80 | Yes |
| | Western part | 0.1934 *<br>(1.8321) | 120 | Yes |

Note: * and *** represent 10% and 1% significance levels, respectively.

It has been found that the impact coefficient of DE is not significantly positive in the eastern region of China, while it is significantly positive in the central and western regions. This indicates that DE has a remarkable influence on promoting HQAD in the central and western regions. The main reason is that, in recent years, the national policy has tilted and strategic layout of the central and western parts of the country has been altered so that the digital economy in the central and western parts of the country has a greater space for development. In the East, due to the earlier start of the digital economy, affected by the "siphon effect", it is in a period of diminishing marginal effect, which leads to its relatively insignificant effect on the promotion of agriculture development. At the same time, it also further demonstrates the necessity and rationality of continuing to promote the growth of digital villages in central and western China. In addition, the fact that the western region does not have as high a level of significance as the central region may be due to the western region having lower development conditions than the central region in all aspects, as well as lacking talents in the digital economy. It is also relatively reverse in regard to such infrastructure and digital technology, and thus has a relatively small driving effect on the high-quality development of agriculture.

### 4.5. Spatial Spillover Effects

Theoretically, the external network characteristics of the digital economy are conducive to the realization of cross-sectoral and cross-regional free flow of agricultural resource elements, and the resulting technological innovation spillover effect will promote the coordinated development of agricultural quality between regions. On this basis, this paper empirically analyzes the spatial spillover effect of DE for HQAD. In this paper, the spatial autocorrelation between DE and HQAD under the economic-geographical nested matrix in each province year is firstly calculated by using the whole-area Moran's I index method, and the results are obtained in Table 9, both of which are significantly positive, which indicates that there is a positive spatial correlation between DE and HQAD in Chinese cities during the sample period.

Further, the spatial distributions of HQAD and DE are examined separately in this paper in Moran scatter plots for two randomly selected years (Figure 2). Among them, Jiangsu (JS), Zhejiang (ZJ) and Shanghai (SH) are all distributed in the first quadrant, indicating that these regions have a better level of development. However, regions such as Yunnan (YN) and Gansu (GS) show a trend of low-low aggregation and slower development. It can be found that most of the cities are distributed in the first quadrant and the third quadrant, and this indicates that there is spatial agglomeration in both HQAD and DE of the cities, which preliminarily verifies the spillover effect of the two.

**Table 9.** Spatial correlation characteristics.

| Year | HQAD | | DE | |
|---|---|---|---|---|
| | **Moran's I** | **Z-Value** | **Moran's I** | **Z-Value** |
| 2012 | 0.4374 *** | 4.4463 | 0.2859 *** | 3.1675 |
| 2013 | 0.4520 *** | 4.5846 | 0.2469 *** | 2.7518 |
| 2014 | 0.4466 *** | 4.5410 | 0.2464 *** | 2.7612 |
| 2015 | 0.4393 *** | 4.3206 | 0.2293 *** | 2.5978 |
| 2016 | 0.4569 *** | 4.6591 | 0.1982 *** | 2.2820 |
| 2017 | 0.4619 *** | 4.7054 | 0.1719 ** | 2.0364 |
| 2018 | 0.4697 *** | 4.7982 | 0.1651 ** | 1.9925 |
| 2019 | 0.4671 *** | 4.8017 | 0.1545 ** | 1.8924 |
| 2020 | 0.4370 *** | 4.5208 | 0.1472 * | 1.8193 |
| 2021 | 0.4194 *** | 4.3206 | 0.1880 ** | 2.2066 |

Note: Moran's index was calculated using stata16.0 software. ***, **, and * denote 1%, 5%, and 10% levels of significance, respectively.

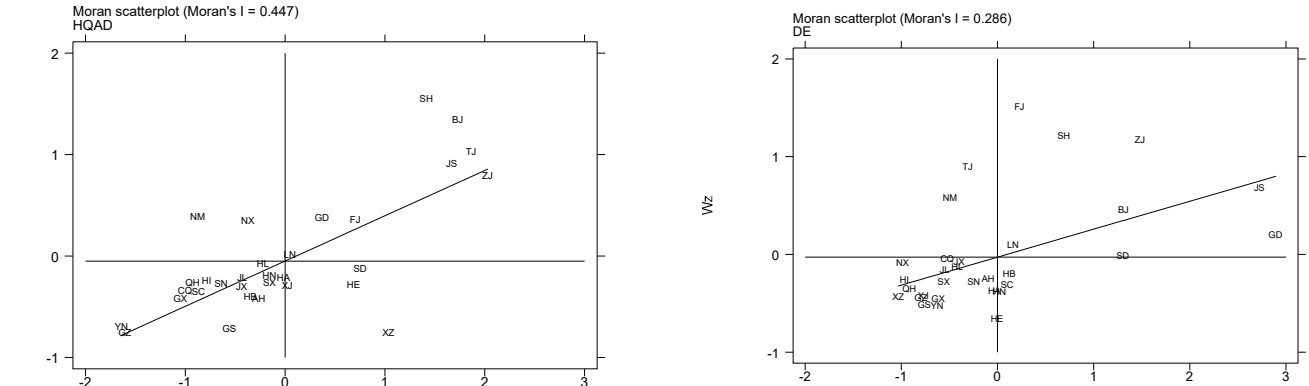

**Figure 2.** Moran's I scatter plot of high-quality agricultural development and digital economy.

In this study, concerning the research findings of Elhorst et al. [45], the most compatible spatial econometric model is screened by the LM test, LR test, and Hausman test, and comparing the size of Log L value, and the comparison shows that the spatial lag (SAR) model is the most compatible. On this basis, the model under the spatial economic geography is a nested matrix for regression analysis, and the conclusions obtained are shown in Table 10. It was found that the coefficients of the spatial autoregression coefficients with DE were positive in the case of nested matrices. This suggests that both HQAD and DE have spatial spillover effects, where HQAD in the neighboring region promotes agricultural development in the region and DE promotes HQAD in the region and the neighboring region. In addition, the study also categorizes the coefficients of digital economy in the spatial lag model into direct effect, indirect effect, and total effect. The research results in Table 10 show that the direct effect, indirect effect, and total effect of DE on HQAD are all positive, among which the direct effect is the most significant. In these two effects, the proportion of direct effect and indirect effect is 79.91% and 20.09% respectively. This suggests that DE has contributed more to HQAD in the region and relatively little to neighboring areas.

**Table 10.** Spatial spillover effects.

| Variable | Spatially Nested Matrix (Math.) |
| --- | --- |
| DE | 0.176 *** (5.77) |
| $\rho$ | 0.199 ** (2.26) |
| Direct effect | 0.179 *** (5.70) |
| Indirect effect | 0.045 * (1.83) |
| Total effect | 0.224 *** (5.22) |
| Control variables | Yes |
| Province fixed effects | Yes |
| $R^2$ | 0.682 |
| Log-likelihood | 822.7976 |

Note: *, ** and *** indicate significance at the 10%, 5% and 1% levels, respectively.

## 5. Conclusions and Policy Implications

In the study, 31 provinces in China from 2012 to 2021 were adopted as research objects, utilizing the entropy evaluation method to research the influence mechanism of the digital economy on the high-quality development of agriculture, which constructs a fixed effect model, a mediation effect model, and a spatial econometric model. The study covered the following main points: First, the digital economy promotes high-quality agricultural development significantly, which still holds a facilitating role after the endogeneity test and robustness test. Second, the digital economy can promote the high-quality development of agriculture through a rationalizing industrial structure, which is specifically reflected in the influence path of "digital economy—rationalization of industrial structure—high-quality development of agriculture." Third, the mechanism of the digital economy's impact on high-quality agricultural development is more significant in the central and western regions and is greater in the central region than in the western region. Finally, there is a spatial spillover effect of the digital economy on the high-quality development of agriculture. The direct effect is greater than the indirect effect, indicating that the pulling effect of the digital economy on the high-quality development of agriculture in the region is more obvious. The pulling effect on the neighboring regions is relatively small.

This study provides the following recommendations to strengthen the agricultural sector and rural areas: Firstly, building a strong rural infrastructure and promoting digital innovation in agriculture; this can be achieved by utilizing the benefits of 5G networks, which offer low latency, high traffic volume, and multiple connections. By building an all-round agricultural and rural cyberspace, rural areas can have a solid digital foundation. It also recommends increasing investment in agricultural research and strengthening the connection between agricultural enterprises and research institutes. This will help to increase the level of agricultural mechanization. Meanwhile, the study suggests cultivating the digital literacy of farmers. This can be achieved by providing training in Internet technology, promoting the use of intelligent agricultural facilities, and carrying out agricultural e-commerce operations. Creating brands of special agricultural products, expanding sales channels, and promoting the digitization of agricultural sales can also help to achieve this goal.

Secondly, promoting the development of digital platforms and easing the path of industrial upgrading. The industrial structure of the agricultural industry plays a crucial role in this process, and it is necessary to pay attention to the obstacles faced by the industry during its development and transformation. By creating a digital platform, the agricultural industry can establish a new type of industry chain that addresses the issues of disconnected and fragile, scattered, and disorderly traditional agricultural operations. This approach can also help to explore the advantages of local characteristic industries, enhance the cohesion of regional resources and technology, and promote the rationalization and high-endization of the operation system and production mode. Ultimately, this will lay a solid foundation for the stable development of advanced agriculture.

Thirdly, there should be an emphasis on developing regional shortcomings and creating differentiated strategies. The eastern region, with its advantages in the economy and transportation, should explore a new model of agricultural development that can provide valuable lessons for the central and western regions. The central region should continue to provide necessary support in terms of talent, technology, and finances. The western region must expedite the improvement of digital infrastructure, eliminate outdated agricultural machinery, reduce the cost of network use in rural areas, and accelerate the entry of network equipment, such as broadband, computers, and smartphones, into rural areas. This will enable agricultural development in less developed regions to enjoy the benefits of the development of the digital economy at an early stage.

Finally, to achieve regional development, it is important to strengthen internal and external linkages while building a synergistic development pattern. This can be achieved by fully utilizing the network effect of digital innovation to strengthen the linkage of agricultural development within the regions. The regions should use the Internet to build multifunctional digital platforms, promote digital innovation, and share agricultural factor resources. This will help in strengthening the linkage of agricultural development within the region and city and create a new situation of complementary advantages and coordinated development. In addition, regions with a high level of digital innovation-enabled high-quality agricultural development should actively share successful experiences and management techniques, promote interregional technical exchanges and cooperation, and assist in the balanced development of modern agriculture in all regions.

## 6. Discussion

The digital economy, as a powerful driving force for economic and social development, offers more possibilities for the sustainable development of agriculture and also brings new opportunities for high-quality agricultural development. First, the digital economy is conducive to agriculture's enhanced development momentum, improved quality, optimized structure, and green transformation. Digital technology can improve the efficiency of agricultural production, provide traceability and quality assurance, provide tools for accurate decision-making and regulation (as well as protect the environment save costs by adjusting the form of production), and promote the sustainable development of the agricultural industry. Second, the digital economy realizes the high-quality development of agriculture through the rationalization of the industrial structure. The digital economy can promote the rationalization of agricultural operation mode and internal and external structure of agriculture, promote the coordination and integration of various rural industries, and promote the reform of the agricultural supply side. Finally, the digital economy brings new elements and combinations to the production system, creating new value and enhancing the information exchange and interaction between regional agricultural market players, making the impact of the digital economy on agricultural development more significant. With the continuous supply of energy for the digital transformation of agriculture, the economic growth of agriculture will be stronger, the size of the market will be further expanded, production methods will be cleaner, and the impact on the process of sustainable development will be more comprehensive and deeper. Under the guidance of the digital economy, producers will demand more from farmers. It is necessary to expand the coverage of networks in rural areas, strengthen the training of digital talents, and promote the integration of the digital economy and agriculture. Only by fully grasping the advantages of digital technology and solving related problems within the agricultural sector can the pace of high-quality agricultural development be accelerated.

## 7. Limitations of this Study

This study discusses the influence and path of the digital economy on the development of high-quality agriculture and hopes that it can be used as a reference for the construction of digital rural areas in China; however, the study still has some limitations. First, we chose the index of the quality of the digital economy and agriculture by reading

high-quality studies; however, considering the availability and complexity of the relevant data, the selected indicators may still not be comprehensive enough. Secondly, there are many factors affecting agricultural development, but this study only selected industrial structure rationalization as a mediating variable without selecting and analyzing other possible mediating variables. Finally, the data of this study is mainly gathered from the China Statistical Database and regards Chinese provinces as the research object, so the conclusions and recommendations may be limited to China and other countries with the same national conditions.

**Author Contributions:** Conceptualization, J.H. and J.Y.; data curation, J.Y.; formal analysis, J.H. and J.Y.; methodology, J.Y.; software, J.Y.; supervision, J.H., Y.S., Q.X. and Y.Z.; writing—original draft, J.Y.; writing—review and editing, J.H., J.Y., Y.S., Q.X. and Y.Z. All authors have read and agreed to the published version of the manuscript.

**Funding:** This research was funded by the National Natural Science Foundation of China, grant number: 72373036; Henan Postdoctoral Research Grant Program, grant number: 2023H126.

**Institutional Review Board Statement:** Not applicable.

**Informed Consent Statement:** Not applicable.

**Data Availability Statement:** Data Availability Statement: The data in this paper are publicly available and mainly came from the National Bureau of Statistics of China: http://www.stats.gov.cn/sj/ (accessed on 7 April 2023).

**Conflicts of Interest:** The authors declare no conflict of interest.

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
