# Peer review of "The Enabling Effect of Digital Economy on High-Quality Agricultural Development-Evidence from China"

_sustainability, doi:10.3390/su16093859_

Round 1

Reviewer 1 Report

Comments and Suggestions for Authors

This study explores the impact and mechanism of the digital economy on the high-quality

development of agriculture by the fixed effect model, the mediation effect model, and the spatial

spillover mode.And I think this topic has certain research significance。The structure of the article is complete and the content is sufficient, but the description is not objective enough, and some details are still worth noting.The details that need to be amended are as follows:

1. The abstract part of this paper describes the research process, method, purpose and conclusion in detail, but the description of the research background is also very little, and it is suggested to add.

2. The introduction describes in detail the research background, the relationship between the importance of digital economy and the high-quality development of agriculture, and the other marginal contributions of this paper, but it lacks a description of the structure of the paper, and it is suggested to add.

3. As for part 2.1 of the paper, the author puts forward the basis for hypothesis 1 from four aspects, but there is no reference mark in the paper, which cannot explain the reliability of the statement, and suggests the author to revise it.

4. When discussing hypothesis 2, it is inappropriate for the author to use agricultural product transportation and marketing to represent the external structure of agriculture, and the paper is not clear about the distinction between the internal and external structure of agriculture, and it is suggested to add.

5. As for the diagram, Figure 1 is incorrectly formatted, please ask the author to make changes.

6. For the rationalization of industrial structure, the author sometimes describes it as the intermediate effect variable, and sometimes describes it as the indirect effect variable. The two are different, and it is suggested that the author make a unified opinion.

7. In this paper, the author set up evaluation indicators from four aspects to discuss the high quality development of agriculture, but the basis for selecting these indicators is not fully explained, and the author is suggested to modify.

8. In the empirical process, the author listed the empirical results in a table, but did not make a clear explanation for each model, so the results could not be completely distinguished from the description, and it is suggested that the author modify it.

9. In the last paragraph of 4.5, when the author describes the table, the table enumerates errors and suggests that amendments be made

10. In the part of conclusion and policy meaning, the author has a situation of sentence confusion, it is suggested that the author carefully check and make changes.

 In a word, this article is accepted in principle, but it is recommended to make significant revisions.

Comments on the Quality of English Language

This study explores the impact and mechanism of the digital economy on the high-quality

development of agriculture by the fixed effect model, the mediation effect model, and the spatial

spillover mode.And I think this topic has certain research significance。The structure of the article is complete and the content is sufficient, but the description is not objective enough, and some details are still worth noting.The details that need to be amended are as follows:

1. The abstract part of this paper describes the research process, method, purpose and conclusion in detail, but the description of the research background is also very little, and it is suggested to add.

2. The introduction describes in detail the research background, the relationship between the importance of digital economy and the high-quality development of agriculture, and the other marginal contributions of this paper, but it lacks a description of the structure of the paper, and it is suggested to add.

3. As for part 2.1 of the paper, the author puts forward the basis for hypothesis 1 from four aspects, but there is no reference mark in the paper, which cannot explain the reliability of the statement, and suggests the author to revise it.

4. When discussing hypothesis 2, it is inappropriate for the author to use agricultural product transportation and marketing to represent the external structure of agriculture, and the paper is not clear about the distinction between the internal and external structure of agriculture, and it is suggested to add.

5. As for the diagram, Figure 1 is incorrectly formatted, please ask the author to make changes.

6. For the rationalization of industrial structure, the author sometimes describes it as the intermediate effect variable, and sometimes describes it as the indirect effect variable. The two are different, and it is suggested that the author make a unified opinion.

7. In this paper, the author set up evaluation indicators from four aspects to discuss the high quality development of agriculture, but the basis for selecting these indicators is not fully explained, and the author is suggested to modify.

8. In the empirical process, the author listed the empirical results in a table, but did not make a clear explanation for each model, so the results could not be completely distinguished from the description, and it is suggested that the author modify it.

9. In the last paragraph of 4.5, when the author describes the table, the table enumerates errors and suggests that amendments be made

10. In the part of conclusion and policy meaning, the author has a situation of sentence confusion, it is suggested that the author carefully check and make changes.

 In a word, this article is accepted in principle, but it is recommended to make significant revisions.

Reviewer 2 Report

Comments and Suggestions for Authors

Abstract: Page 1, Line 5: One could also say “…by the fixed effect, mediation effect, and the spatial spillover models.” Remove line “The study draws the following conclusions”. In lieu of “Overall” suggest to replace with “In summary”.

Introduction: As this section is covering both Intro and Lit Review, wouldn’t it be more prudent to include the subtitle as “Introduction and Literature Review”. Please define “High-Quality Agriculture” either in abstract or in first few sentences of Introduction; even if it’s referencing Xia Xianli or Wang Xingguo’s definition from later in the Introduction (or move that paragraph closer to the beginning of introduction). Page 2, midway down should be “relationship” rather than relation. The line  2/3 of the way down starting with “Weak digital transformation…” is currently a sentence fragment. It should not be “The Ozili” but “Ozili” found…Page 3, near end of Introduction, it should be “Third from…” (not capitalizing “From”).  I found this section to be concise and well-encapsulated.

Theoretical Analysis and Research Hypothesis: Looks good.

Methodology and Data: Page 6, under dependent variable discussion, is there a reason that entropy method was used to measure the weights of each index. Tables should not be split between different pages. Page 8, Table 3, mediating variable Theil index should be spelled “Thiel”.

Empirical Testing and Discussion: Is the decision to use the Fixed Effects model solely based on the Hausman test results for the various model tests?

Discussion: The first sentence “Sustainable development is one of the key goals of the current social development” is an awkward statement, would suggest rewriting it.  Midway in first paragraph, green development is included with other factors of benefits of the digital economy, however, green development wasn’t touched on in the paper.

Might be good to separate last paragraph into a subsection “Limitations of this study”.

Conclusions and Policy Implications: Page 16, under the Third paragraph, there’s a stray sentence “The eastern region should.” Also, incomplete sentence “On the other hand, promotes interregional technical exchanges and cooperation, and promotes the balanced development of modern agriculture in all regions”.

Overall comments: No manuscript line numbering. The majority of the paper is well written and probably needs 1-2 more turns through it for grammatical precision. It’s well organized, with some repetition of the findings relating to the hypotheses, which is a positive for the reader.

Comments on the Quality of English Language

Still needs another check for some minor grammatical improvements.
